# Identification of Potential Biomarkers for Diagnosis of Patients with Methamphetamine Use Disorder

**DOI:** 10.3390/ijms24108672

**Published:** 2023-05-12

**Authors:** Won-Jun Jang, Sang-Hoon Song, Taekwon Son, Jung Woo Bae, Sooyeun Lee, Chul-Ho Jeong

**Affiliations:** 1College of Pharmacy, Keimyung University, 1095 Dalgubeoldaero, Dalseo-gu, Daegu 42601, Republic of Korea; mrdoin76@kmu.ac.kr (W.-J.J.); gnsdl0330@naver.com (S.-H.S.); jwbae11@kmu.ac.kr (J.W.B.); 2Korea Brain Bank, Korea Brain Research Institute, Daegu 41062, Republic of Korea; taekwon@kbri.re.kr

**Keywords:** methamphetamine, prediction model, methamphetamine use disorder, RNA sequencing, peripheral biomarker

## Abstract

The current method for diagnosing methamphetamine use disorder (MUD) relies on self-reports and interviews with psychiatrists, which lack scientific rigor. This highlights the need for novel biomarkers to accurately diagnose MUD. In this study, we identified transcriptome biomarkers using hair follicles and proposed a diagnostic model for monitoring the MUD treatment process. We performed RNA sequencing analysis on hair follicle cells from healthy controls and former and current MUD patients who had been detained in the past for illegal use of methamphetamine (MA). We selected candidate genes for monitoring MUD patients by performing multivariate analysis methods, such as PCA and PLS-DA, and PPI network analysis. We developed a two-stage diagnostic model using multivariate ROC analysis based on the PLS-DA method. We constructed a two-step prediction model for MUD diagnosis using multivariate ROC analysis, including 10 biomarkers. The first step model, which distinguishes non-recovered patients from others, showed very high accuracy (prediction accuracy, 98.7%). The second step model, which distinguishes almost-recovered patients from healthy controls, showed high accuracy (prediction accuracy, 81.3%). This study is the first report to use hair follicles of MUD patients and to develop a MUD prediction model based on transcriptomic biomarkers, which offers a potential solution to improve the accuracy of MUD diagnosis and may lead to the development of better pharmacological treatments for the disorder in the future.

## 1. Introduction

Drug use disorder (DUD) is a global health problem that poses a threat to individuals and society. Typically, it begins with drug-seeking or drug-taking behavior and progresses into a chronic and relapsing disorder characterized by compulsive drug seeking and increased drug intake [1]. During the transition from casual to compulsive drug use, a combination of positive and negative reinforcement can motivate compulsive drug use, despite the risks and awareness of adverse events. Repeated drug use can impair the immune system and cause neurobiological changes in the reward circuitry of the brain, leading to physical dependence, tolerance, withdrawal upon discontinuation, craving, and relapse [1,2,3].

Methamphetamine (MA) is a highly addictive psychostimulant that frequently leads to physical and psychological dependence by disrupting the central nervous system (CNS). MA use disorder can alter the release and activity of various neurotransmitters, including dopamine, glutamate, norepinephrine, and serotonin, in the CNS, leading to significant clinical withdrawal symptoms such as depression, psychosis, suicidal ideation, hypersomnia, irritability, anxiety, and intense craving for MA [4,5]. There are currently no FDA-approved medications available that can effectively reduce drug use in people addicted to MA without the possibility of relapse. The focus of recent studies regarding the pharmacological treatment of methamphetamine use disorder (MUD) has been on developing therapeutic drugs that target addiction-related neurological systems. Various medications such as dexamphetamine [6], methylphenidate [7], and naltrexone [8] are being investigated to improve their capacity to target the dopaminergic, serotonergic, GABAergic, and/or glutamatergic pathways. Additionally, non-pharmacological treatments such as8 cognitive behavioral therapy [9], motivational interviewing [10], and vaccines [11] that prevent drugs from entering the brain can also be considered as treatment options for MUD patients.

Effective treatment of MUD requires an accurate diagnosis of the addiction state of MA users. Currently, the clinical diagnosis of MUD relies mainly on self-reports of the patient’s drug-seeking behavior and psychological conditions, which can be problematic for patients with less severe symptoms or those who are reluctant to seek treatment. Thus, a more objective method that incorporates underlying neurobiological factors of MA use disorder is needed. In previous studies, neuroimaging was used to evaluate MA use disorder or withdrawal in patients by measuring neuronal markers such as N-acetyl aspartate and glutamate [12,13,14]. In addition, the severity of MA use disorder has been evaluated through hair analysis based on the concentrations of MA and its main metabolite, amphetamine (AM) [9,15]. However, hair metabolic analysis has limitations, as lower levels of MA and AM in hair do not necessarily indicate a lower severity of MUD. Therefore, additional diagnostic analysis is needed to overcome the shortcomings of hair metabolic analysis.

Efficient diagnosis of MUD requires biomarkers that cover diverse neurobehavioral and pathological symptoms of MA use disorder. MUD is a chronic, relapsing brain disease that is caused by changes in neuronal adaptation resulting from MA use [16,17,18]. These changes indicate that the brain contains information about an individual’s current disease status and prognosis. Therefore, identifying reliable biomarkers in the brain that reflect pathological changes caused by MA use disorder can aid in the monitoring of treatments [19,20]. In previous studies, we investigated the role of common and differential alterations of expression of several genes and pathways in the striatum and peripheral whisker follicles of MA-administered rats for this purpose [21,22,23]. In the present study, we aimed to study and identify reliable biomarkers in hair follicle cells of humans addicted to MA. The results of this study will provide important information for diagnosing the state of MUD and will aid in developing better pharmacological treatments in the future.

### 1.1. Contribution

This study is a novel approach to identify non-invasive transcriptome biomarkers using human hair follicles for the accurate diagnosis of MUD.A two-step predictive model that can monitor and identify the progression of MUD treatment was developed and validated.The proposed biomarkers and predictive model have the potential to improve the accuracy of expert diagnoses and provide better treatment options for MUD patients.

### 1.2. Literature Review

Recent papers present new methods for the diagnosis and treatment of MA use disorder and emphasize the importance of microRNAs in early disease diagnosis [24]. The authors investigate miR-222 as a potential biological marker for diagnosing MA use disorder. The study found that miR-222 expression was significantly increased in patients with the disorder, suggesting its use in patient diagnosis. Chand et al. [25] conducted a study investigating the role of microRNA-29a in chronic MA use disorder. The study found reduced levels of microRNA-29a in rats with chronic MA use disorder. The results suggest that microRNA-29a may interact with addiction-related neurotransmitters and regulate addictive behavior. This research expands our understanding of drug use disorder and provides new ideas for treatment and prevention approaches.

Several up-to-date reports on the diagnosis of other diseases are available. Shabbir et al. [26] propose a non-invasive approach for early prediction of malignant mesothelioma using data mining and machine learning techniques. The proposed method involves collecting data through non-invasive tests such as blood tests from patients and building a prediction model using machine learning algorithms. The results demonstrate high accuracy in early prediction of malignant mesothelioma, which could improve survival rates through early diagnosis and treatment. Another study offers new possibilities in the field of cancer diagnosis and prevention. Alam et al. [27] applied machine learning techniques such as Random Forest, Support Vector Machine, and Artificial Neural Network to identify the causes of malignant mesothelioma. The study analyzed various attributes such as age, gender, residential city, and asbestos exposure to understand the causes of malignant mesothelioma. Although conducted on an imbalanced dataset, the results indicate the potential for machine learning techniques to be used in cancer diagnosis and prevention.

## 2. Results

### 2.1. RNA-seq Data Processing

Our previous studies have shown that there was no correlation between the psychiatric diagnosis results and the concentrations of hair MA among different MUD patient groups, including MUD patients who have been treated in the past (former patients; FPs) and MUD patients who are currently being treated (current patients; CPs) [28,29]. Therefore, we need a validated MUD prediction model that can accurately determine the addiction state of MA users. In this study, we initially recruited 85 participants, including 30 healthy controls (HCs), 33 CPs, and 22 FPs. However, samples of 2 HCs, 10 CPs, and 11 FPs had to be excluded from further RNA-seq analyses because of poor RNA sample quality. Therefore, downstream analysis proceeded with 57 samples, including HCs (*n* = 26), CPs (*n* = 20), and FPs (*n* = 11) (Figure 1).

To evaluate the differences and/or similarities between the RNA transcriptomes of the three groups, we performed principal component analysis (PCA) [30] and hierarchical cluster analysis (HCA) [31] on data from a total of 14,358 genes and visualized the results using MetaboAnalyst 5.0 (https://www.metaboanalyst.ca/, (accessed on 9 May 2023)) (Figure 2A–C). We recognized that multivariate analysis methods are suitable for assessing relationships and influences between multiple variables, as well as for understanding patterns and structures in the data [32]. Among these methods, we first examined the overall correlation of the data using PCA, which can analyze useful correlations between data samples without considering relationships between input variables through dimensionality reduction [33]. The differentially expressed genes (DEGs) of these groups (HC, CP, and FP) were clustered based on the quantitative results of transcripts following the PCA analysis. The PCA and HCA revealed that the HC group was clearly clustered compared with the other groups, but the CP and FP groups appeared to be mixed with each other without showing clustering (Figure 2B,C). As expected, we confirmed that the clustering results through multivariate analysis were not consistent with the results of the self-questionnaire. Therefore, we tried to reinterpret the clustered results based only on gene expression patterns to exclude the uncertainty of the self-questionnaire results. As a result, we realized that the two patient groups can be re-classified into new clusters located in the first quadrants of the two-dimensional plane (Q1), away from the HCs (NR patients, non-recovered patients, *n* = 9), and clusters located in the second and third quadrants of the two-dimensional plane (Q2–Q3) areas, close to the HCs (AR patients, almost-recovered patients, *n* = 22) (Figure 2B,C). Consequently, we will evaluate the next analysis using the newly clustered groups based on the expression of transcripts.

### 2.2. Identification of DEGs Using Multivariate Analysis and Keg Pathway Enrichment Analysis

A total of 6106 DEGs were identified from 14,358 genes (false discovery rate (FDR) < 0.05, fold change (FC) ≥ |1.5|) (Appendix A). Figure 2D,E show that 5683 DEGs were found, with 2657 up-regulated genes and 3026 down-regulated genes in the NR group compared with the HC group (Appendix A). Additionally, 5138 DEGs were identified, with 2938 up-regulated genes and 2200 down-regulated genes in the AR group compared with the NR group (Appendix A). Finally, 268 DEGs were dysregulated, with 181 up-regulated genes and 87 down-regulated genes in the AR group compared with the HC group (Appendix A). It was observed that the NR group exhibited a difference of 93.1% (5683 DEGs) and 84.2% (5138 DEGs) of the total DEGs compared with the HC and AR groups, respectively. This suggests that the NR group has a distinct gene expression pattern compared with the HC and AR groups. On the other hand, the AR group showed a difference of 268 DEGs, which is only 4.4% of the total DEGs compared with that of the HC group. However, the AR group exhibited a difference of 5138 DEGs, which is 84.2% of the total DEGs compared with that of the NR group. Therefore, it was concluded that the gene expression pattern of the AR group is similar to that of the HC group, but not to that of the NR group. Based on this, a state transition circuit for MUD patients was proposed among the three groups: (1) the addiction state (NR; group of patients requiring continuous care), (2) the state of transition from drug use disorder to protracted abstinence (AR; group of patients returning to a normal state), and (3) the normal state (HC; group of healthy controls).

We performed KEGG pathway enrichment analysis to investigate the functional implications of the DEGs identified in MUD patients during state transitions. Enrichment analysis revealed several KEGG pathways that were significantly enriched at each of the transition states. At the transition state of ‘MA use disorder’ (HC to NR), we observed significant enrichment of 15 up- and 30 down-regulated pathways of genes (FDR < 0.05) (Appendix A). These pathways were ranked by the number of genes included in the pathway (Appendix A). At this transition state, the top ranks of pathways containing up-regulated DEGs were associated with virus infection and neurological diseases such as herpes simplex virus 1 infection, Alzheimer’s disease, Huntington’s disease, amyotrophic lateral sclerosis, Parkinson’s disease, and prion disease. Additionally, the top ranks of pathways containing down-regulated DEGs were related to cancer, signaling pathways including PI3K-Akt and MAPK, and axon guidance.

At the transition state of AR to HC, 181 DEGs were up-regulated, while 87 DEGs were down-regulated. However, we did not observe any significant pathways containing these DEGs. This finding suggests that the AR group has no significant functional difference compared with the HC group. We observed significant changes in enriched pathways between the transition state from HC to NR and the transition state from NR to AR. More specifically, 13 out of the total 15 pathways that included up-regulated DEGs in the NR group were switched to pathways including down-regulated DEGs in the AR group (Appendix A). Similarly, 27 out of the 30 pathways including down-regulated DEGs in the NR group were switched to pathways including up-regulated DEGs in the AR group (Appendix A). Additionally, 2033 and 2697 DEGs that were up- or down-regulated in the NR group were reversely switched to the down- and up-regulated DEGs in the AR group, respectively (Appendix A). These results suggest that the expression pattern of DEGs dramatically changes according to the transition of three states, NR, AR, and HC.

### 2.3. Partial Least Squares-Discriminant Analysis (PLS-DA)

We conducted integrative analysis of PLS-DA and network analysis to identify key genes that can distinguish the state of MUD, following the procedure shown in Figure 3A. To confirm the differentiation and similarity among the three groups, we performed PCA on 6106 DEGs. As shown in Figure 3B, the first two principal components (PCs) explained 56.0% of the expression variances, with PC1 (37.4%) clearly separating the NR group from the other groups, and PC2 (18.6%) showing no clear distinction between groups, indicating a similar pattern to the PCA result analyzed using total DEGs. Next, we employed the PLS-DA method to identify the optimal components that demonstrate similarities and dissimilarities between groups and construct a prediction model. PLS-DA enables the analysis of interactions among input variables while addressing the issue of multicollinearity that may diminish the reliability and predictive capability of the analysis [34]. This method offers benefits in classification modeling with limited samples and in evaluating the significance of transcriptome components through dimensionality reduction [35]. In the PLS-DA analysis, a clear separation among the three groups was observed, with component one and component two accounting for 29.1% and 15.0% of the variables, respectively, and 44.1% of variables in total. The HC, NR, and AR groups were in the Q3, Q4, and Q1 area, respectively, indicating that the changes in gene expression pattern are expected to switch in the order of (a)–(b) during MUD treatment (Figure 3C).

To identify key genes, we quantified the contribution of each component in components one and two of PLS-DA as a variable importance in projection (VIP) score, and selected DEGs with a VIP score greater than 1.0 in each component (Appendix A). The number of DEGs with a VIP score > 1.0 in components one and two was 847 and 1018, respectively. The Venn diagram in Figure 3D shows that the number of DEGs included only in components one and two was 273 and 444, respectively, and the number of common DEGs was 574.

In summary, we constructed a clustering model for the three groups using the PLS-DA method and assessed components one and two, which best reflected the characteristics of each group, using the VIP score. We then screened the 574 most influential transcripts.

### 2.4. Construction of Protein-Protein Interaction (PPI) Network and Network Centrality Analysis

To identify functionally significant DEGs and understand their biological roles, we performed PPI network analysis [36] on the selected 1291 DEGs. From the STRING database, we constructed a PPI network consisting of 741 nodes using these DEGs (Figure 3E). We then used network centrality analysis to select genes that were topologically important within the network, based on their node degree and betweenness centrality. Genes whose values were above average within the network were selected, resulting in 149 genes being chosen (Figure 3F). The averages of node degree and betweenness centrality were DC = 4.48 and BC = 0.0064, respectively. The list of the 149 genes selected through network and centrality analysis can be found in Appendix A.

### 2.5. Functional Annotation Analysis

To gain a better understanding of the biological functions of the selected 149 DEGs, we utilized the DAVID (Database for Annotation, Visualization, and Integrated Discovery) bioinformatics resources to perform functional annotation clustering. As a result, 39 genes were identified as belonging to five clustering annotation groups, with an enrichment score of >1.5 and a similarity term overlap of >5. The KEGG pathway with significance levels set at FDR < 0.05 in each cluster is shown in the diagram in Figure 4A. The majority of the genes are associated with vascular smooth muscle contraction and the oxytocin signaling pathway in the first cluster with an enrichment score of 3.13, and in the second cluster, long-term depression with an enrichment score of 2.47.

Additionally, dopaminergic synapse in the third cluster (enrichment score 2.23) and pathways of neurodegeneration-multiple diseases in the last cluster (enrichment score 1.52) are noteworthy. The lists of particular genes for the most enriched clusters are shown in Figure 4B–F, and the annotation clusters are listed in Appendix A.

### 2.6. Construction of MUD Prediction Models

ROC curve analysis is widely used in biomedical fields as a standard method for evaluating biomarker performance [37].

In this study, we performed ROC curve analysis to identify the most important genes among 39 candidate genes. To select reliable biomarkers that represent each state of MUD treatment, we employed a two-step strategy (Figure 5A). First, we selected biomarkers to distinguish the NR group from the other groups. Then, we selected biomarkers to distinguish between the AR and HC groups. We constructed six ROC models between the NR and other groups, and all models showed an AUC value of one, indicating that the selected biomarkers can clearly distinguish the NR and other groups (Figure 5B). We then performed prediction analysis for both groups, which confirmed that the NR and other groups were accurately predicted with prediction accuracy >98% (Figure 5C,D and Appendix A). Five genes (proteasome 20S subunit alpha 2; *PSMA2*, Rac family small GTPase 3; *RAC3*, protein phosphatase 1 regulatory subunit 12A; *PPP1R12A*, dishevelled segment polarity protein 1; *DVL1*, SUFU negative regulator of hedgehog signaling; *SUFU*) were finally selected as biomarkers to distinguish NR from other groups (Figure 5H, Appendix A).

Next, we performed multivariate ROC analysis between the AR and HC groups. Six models were constructed, and model three was adopted with the fewest number of genes among the models with an AUC > 0.9 (AUC = 0.907, CI = 0.792–1.000) (Figure 5E). Prediction analysis revealed that AR and HC groups were accurately predicted by model three with 81.3% prediction accuracy (Figure 5F,G, and Appendix A). Therefore, five genes (APC regulator of WNT signaling pathway 2; *APC2*, kinesin light chain 3; *KLC3*, NDUFA4 mitochondrial complex associated; *NDUFA4*, Fas associated via death domain; *FADD*, apolipoprotein E; *APOE*) were selected as biomarkers to distinguish AR and HC groups (Figure 5I, Appendix A).

### 2.7. Validation of MUD Prediction Models

A two-step modeling approach was used to test the potential predictability of 10 biomarkers as MUD treatment biomarkers (Figure 6A). The verification step was performed on five test samples that were originally diagnosed as either HC (two samples) or CP (three samples). The ROC curve analysis using the combination of five biomarkers (*PSMA2*, *RAC3*, *PPP1R12A*, *DVL1*, *SUFU*) showed an excellent ability to distinguish NR from other groups (AUC = 1, CI = 1–1) (Figure 6B). Two of the three samples that were originally diagnosed with the CP group were predicted to be NR group in the first step of modeling (Figure 6C). In addition, the PCA using the five test samples and the fifty-seven samples used for modeling showed that two out of the three CP samples belong to the NR group as predicted by the model (Figure 6D).

In the second step of modeling, the combination of five biomarkers (*APC2*, *KLC3*, *NDUFA4*, *FADD*, *APOE*) showed a good ability to distinguish AR from the HC group (AUC = 0.891, CI = 0.74–1) (Figure 6E). Two samples originally diagnosed with the HC group were correctly predicted as HC group, and the remaining CP sample was predicted as the HC group, which is consistent with the result of the PCA (Figure 6F,G). The modeling results indicate that both HC samples were correctly predicted to be HC groups, and three CP samples were finally predicted as two NR groups and one HC group.

A regulatory network based on the functions of the 10 biomarkers is shown in Figure 6H. The biomarkers used in the MUD prediction model are primarily involved in the pathways of cancer, Alzheimer’s disease, and neurodegeneration. The use of these biomarkers confirmed that the MUD prediction model accurately identifies the state of MUD patients recovering from MA use disorder.

## 3. Discussion

There are several screening instruments that have been developed for the clinical diagnosis of DUD. Commonly used drug use disorder screening instruments include the National Institution of Drug Abuse-alcohol, smoking and substance involvement screening test (NIDA-ASSIST), DUD identification test (DUDIT), and drug abuse screening test (DAST). However, the reliability and validity of these instruments have not been fully established, and they are often used in conjunction with quantitative analysis of drugs and/or metabolites in peripheral biospecimens. Although combining drug use disorder screening test scores with analytical results has been shown to improve diagnostic performance in some cases [38,39], their efficacy has not been fully demonstrated, particularly in the diagnosis of MUD. Our previous studies have shown that there is no correlation between MA concentration in hair and drug use disorder screening test scores in current and former patients [28], indicating that hair analysis cannot monitor the recovery status of MUD patients. Therefore, it is urgent to develop a diagnostic method that uses genetic biomarkers to accurately diagnose the state of MUD and predict treatment response.

This work aimed to explore hair follicle gene expression among human subject groups diagnosed with MUD and healthy controls using integrative RNA sequencing data. Traditional diagnostic methods for MUD such as self-reports, interviews with psychiatrists, and metabolic analyses of hair do not show consistent results in diagnosing MUD patients [28]. Based on gene expression in hair follicles, our multivariate analysis showed that the previous classification of patients (FP, CP) under existing diagnostic criteria was reorganized into non-recovered (NR) and almost recovered (AR) patient groups. In this study, we developed a MUD prediction model based on gene expression in hair follicle cells. This model can identify 10 specific biomarkers that differentiate between HCs, NR patients, and AR patients.

Two important genes among these biomarkers that reflect MA use disorder are *PSM2* and *RAC3*. *PSMA2*, a subunit of the 20S proteasome complex involved in the degradation of intracellular proteins, is upregulated in the NR group. Dysfunction of *PSMA2* may contribute to neuronal dysfunction [40] and neurodegenerative diseases such as Parkinson’s disease [41]. Previous reports have identified *PSMA2* as a potential peripheral biomarker for the early diagnosis of Parkinson’s disease [42]. On the other hand, *RAC3*, a member of the Rho family of small GTPases, is downregulated in the NR group. *RAC3* has been implicated in regulating various cellular processes, including axon growth and branching [43], dendritic spine formation [44], and synaptic plasticity [45]. In the nervous system, *RAC3* is involved in the development and function of GABAergic interneurons in the cortex and hippocampus [46]. These findings suggest that *RAC3* may serve as a therapeutic target for neurological disorders that involve dysfunction of GABAergic interneurons, such as epilepsy and schizophrenia. Although the specific roles of these genes and other marker genes in MUD are not yet clear, their dysregulation could contribute to the pathogenesis of MUD by affecting neuronal development, function, and survival. However, further research is needed to fully understand the mechanisms by which these genes contribute to the development of MUD.

In order to test the accuracy of this model, we applied it to three patients who were classified as CP based on the existing diagnostic criteria. The model predicted that two of these patients were NR patients, indicating that they had not fully recovered from MUD. However, the third patient was predicted to belong to the healthy control (HC) group, indicating that they did not exhibit the gene expression patterns associated with MUD. There are several potential explanations for this result. One possibility is that the existing diagnostic criteria may have missed the diagnosis of MUD in the third patient, as traditional diagnostic methods such as self-reports, interviews with psychiatrists, and metabolic analyses of hair can have limitations in identifying MUD patients. Alternatively, the third patient may have previously been diagnosed with MUD but has since fully recovered, leading to a change in their gene expression patterns. In order to validate the accuracy and reliability of our MUD prediction model and the identified biomarkers, further research is necessary using larger patient cohorts.

Identifying peripheral genetic biomarkers in humans addicted to MA with different periods of abstinence is a crucial area of research that can have several advantages. One primary advantage of this study is that it was conducted on humans with drug use disorder rather than an animal model, providing practical gene expression information according to the abstinence period of the people addicted to MA. The identified biomarkers may aid in the early detection and diagnosis of MA use disorder, predict treatment response, monitor recovery progress, and improve our understanding of the biological mechanisms underlying MUD. A secondary advantage of this study is the use of non-invasive hair follicles as a specimen to investigate different states of MUD. Previous studies have suggested that hair follicles can reflect the state of the CNS, in part [21,22,47]. This is the first report to investigate the alteration of gene expression in humans addicted to MA with different abstinence periods using hair follicles. The use of peripheral genetic biomarkers can provide more reliable measurements than conventional diagnostic methods and may reduce the stigma associated with misdiagnosis of drug use disorder. Overall, the identification of peripheral genetic biomarkers in humans addicted to MA with different periods of abstinence has the potential to lead to improved diagnosis, treatment, and understanding of MUD. This study highlights the importance of using human samples and non-invasive methods to identify biomarkers and the potential benefits of such an approach in improving our understanding of drug use disorder and developing better treatments. 

Although the current study has many advantages, there are still some limitations. One of the most significant limitations is the difficulty in obtaining a sufficient number of MUD patient samples due to the specificity of the disease and its association to crime. However, despite the small sample size, the study’s results were meaningful and validated through comprehensive consideration of rigorous sample selection and quality control for RNA-seq. The relatively small sample size problem in the study could be addressed by planning and validating larger human studies in the future. Additionally, it is important to note that genetic biomarkers alone cannot diagnose MUD, and a comprehensive evaluation of the patient’s symptoms and history, along with other diagnostic tools, is necessary for accurate diagnosis.

## 4. Materials and Methods

### 4.1. Participants

The procedures were conducted in compliance with the guidelines of the Declaration of Helsinki on biomedical research involving human subjects and were approved by the Institutional Review Board of Bugok National Hospital (Bugok-myeon, Changnyeong-gun,, Gyeongsangnam-do, Republic of Korea, approval number: BNH-2018-03, approval date: 26 April 2018). The study recruited a total of 85 male participants, including 30 healthy controls and 55 patients with MUD who had been detained for illegal use of MA in the past, from Bugok National Hospital, a national drug use disorder treatment hospital in Korea. All participants provided written informed consent and underwent a comprehensive interview to exclude any significant medical conditions or substance use disorders other than MA use in the past or present. The 55 MUD patients were categorized as 33 current patients with MUD (CPs) or 22 former patients with MUD (FPs).

### 4.2. Sample Collection and RNA Extraction

Hair samples were collected from the posterior vertex area of each subject. About 30 hair strands were obtained by pulling as close to the scalp as possible using hair tweezers. Hair strands with follicles were selected after checking each hair for hair follicle attachment. The hair samples were cut approximately 5 mm from the follicles and stored in envelopes at −80 °C. The hair follicles were preserved using RNA later solution (Qiagen, ON, Canada) for future batch analysis. To extract total RNA from hair follicles, each hair strand was cut into 10–11 fragments of 0.5 mm each using a microsurgical knife. Total RNA was extracted from hair follicle samples of all subjects using TRIzol RNA Isolation Reagents (Life technologies).

### 4.3. RNA Sequencing and Data Processing

RNA samples were extracted from hair follicles of 33 CPs, 22 FPs, and 30 HCs, followed by quality control (QC) for RNA sequencing. The extracted RNA samples underwent QC analysis to assess their eligibility for RNA sequencing. The QC criteria were set at 28S:18S < 1 and RIN < 7. Based on the 62 samples that passed quality control, CP (*n* = 23), FP (*n* = 11), and HC (*n* = 28) were included, while the other 23 samples were excluded from RNA sequencing. Appendix A presents the characteristic information of the 3 groups. The extracted total RNA was used to prepare the mRNA-sequencing library using a TruSeq Stranded mRNA Library Prep Kit (Illumina, San Diego, CA, USA), and the libraries of 62 independent samples were quantified using qPCR and qualified using an Agilent Technologies 2100 Bioanalyzer. For low RIN samples, RNA-seq libraries were generated using the Illumina TruSeq Stranded Total RNA sample preparation kit. All samples were sequenced on a NextSeq 500 (Illumina), and the raw image data were transformed into sequence data using base-calling and stored in FASTQ format. The trimmed reads were aligned to the GRCh38 human reference genome using STAR (version 2.7.9a) [48] and gene-level read counts were generated using the featureCounts function from the Subread package (version 2.0.3) [49]. The presence of significant differential expression was determined using DESeq2 (version 3.10) [50] at the gene level. Raw count data were obtained for the 62 samples through data processing. Finally, the remaining 57 samples (HC = 26, CP = 20, FP = 11) were used for transcriptome profiling. To minimize individual variations of gene expression among the 57 samples, raw count data were normalized using the trimmed mean of M-values (TMM) normalization method in NetworkAnalyst 3.0, a Web-based tool for comprehensive profiling of gene expression data, and normalized data containing 14,358 genes are presented in Appendix A.

### 4.4. Differential Gene Expression Analysis

To determine the presence of statistically significant differential expression, we used NetworkAnalyst 3.0. Next, we selected genes based on their average false discovery rates (FDRs) and average log2-fold expression changes among the three groups (HC, CP, and FP). In making these comparisons, we considered genes to be differentially expressed if their FDR was below 0.05 and the absolute value of their fold change was greater than |1.5|.

### 4.5. Principal Component Analysis and Partial Least Square Discriminant Analysis

Multivariate statistical analyses such as principal component analysis (PCA) and partial least squares-discriminant analysis (PLS-DA) were performed using MetaboAnalyst 5.0 (https://www.metaboanalyst.ca/, (accessed on 9 May 2023)). Initially, we applied PCA to detect trends and cluster transcripts in an unsupervised manner. Next, we used PLS-DA to better identify clustering and screen high-importance transcripts. We used the Variable Importance in Projection (VIP) score to select transcripts that reflected differentiated features in each axis of PLS-DA. We selected transcripts with a VIP cutoff > 1.0. Cross-validation and accuracy tests for the results were evaluated to provide evidence of their reliability (Appendix A).

### 4.6. KEGG Pathway Enrichment Analysis

For functional and pathway enrichment analysis of the DEGs between the groups, Kyoto Encyclopedia of Genes and Genomes (KEGG, https://www.genome.jp/kegg/, (accessed on 9 May 2023)) was performed. The Database for Annotation, Visualization, and Integrated Discovery (DAVID, Web-based online tool (https://david.ncifcrf.gov, (accessed on 9 May 2023)) was used to perform the KEGG pathway enrichment analysis.

### 4.7. Protein–Protein Interaction Network Analysis

For topological analysis, a protein–protein interaction (PPI) network of DEGs was constructed using the cytoscape software (version 3.9.0, an open-source platform for network analysis and visualization). Based on the degree centrality (DC) and betweenness centrality (BC) of the node in the PPI network, significant hub genes of the gene network were selected. The criteria for selecting the BC and DC values were based on values higher than the average value of all nodes included in the network.

### 4.8. Construction and Validation of MUD Prediction Model

The biomarker analysis tool MetaboAnalyst 5.0 was used to perform classical and multivariate receiver operating characteristic (ROC) curve analysis. Normalized transcript values were scaled using a unit variance scaling method for ROC curve analysis, and multivariate ROC curve analysis was used to identify several biomarkers and evaluate the classification performance of the generated models. To select and cross-validate the most important features, a well-established algorithm was used for multivariate ROC curve analysis, and the generated models were validated through 1000 permutation tests using AUC as a performance measure. Univariate ROC analysis was used to evaluate the diagnostic power and its proportions for each gene, which was calculated using the area under the curve (AUC) and the confidence interval (CI). According to the criteria of Jones and Athanasiou [51], AUC values were interpreted as “excellent”, “very good”, “good”, and “reasonable” if they were greater than 0.97, between 0.93 and 0.96, between 0.75 and 0.92, and between 0.6 and 0.74, respectively. Five test samples remaining from RNA-Seq analysis were used to validate the selected biomarkers for diagnosing MUD patients, and a two-step screening method was applied to confirm the status of MUD patients. Cross-validation and accuracy tests for the results were evaluated to provide evidence of their reliability (Appendix A).

## Figures and Tables

**Figure 1 ijms-24-08672-f001:**
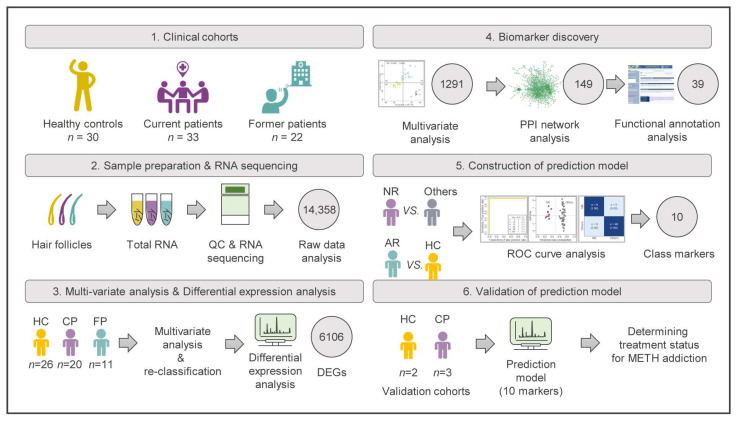
Workflow of the biomarker identification strategy for the diagnosis of MUD patients. Hair follicle samples from 55 MUD patients and 30 matched healthy controls were profiled using RNA sequencing-based transcriptomics. NR: non-recovered patients requiring continuous care; AR: almost-recovered patients recovering to normal; DEGs: differentially expressed genes.

**Figure 2 ijms-24-08672-f002:**
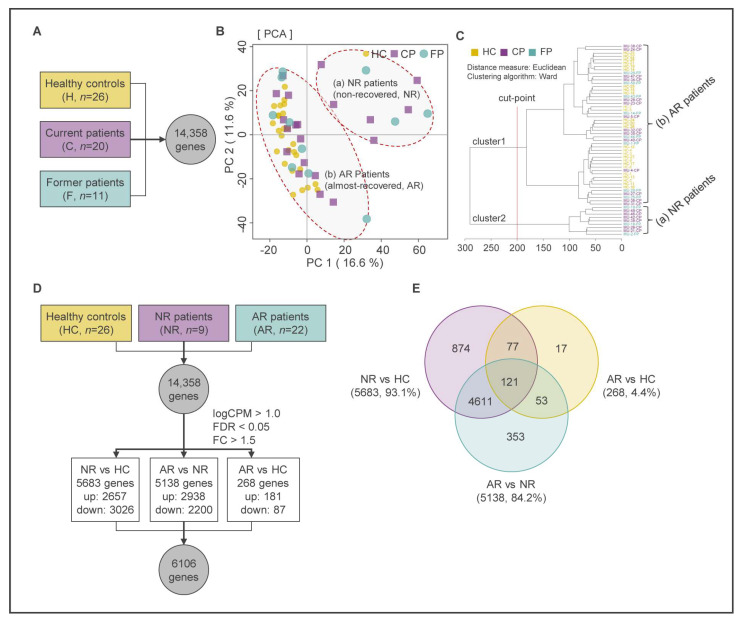
Identification of DEGs using multivariate analysis. (**A**) Extraction of total genes from CP, FP, and HC groups using RNA sequencing analysis. (**B**) PCA score plot of the first two principal components for gene expression levels from samples of MUD patients (CPs and FPs) and HCs. Two large clusters (NR and AR) are formed from three sample groups. (**C**) Dendrogram of hierarchical clustering of three groups based on the Euclidean distance and Ward clustering algorithm. (**D**) Re-classification from the original groups (CP, FP, and HC) based on multivariate analysis. Identification of DEGs from RNA-seq data of re-classified MUD groups (NR, AR, and HC). (**E**) Venn diagram of significantly changed transcripts among NR, AR, and HC. NR: non-recovered patients; AR: almost-recovered patients; HC: healthy control.

**Figure 3 ijms-24-08672-f003:**
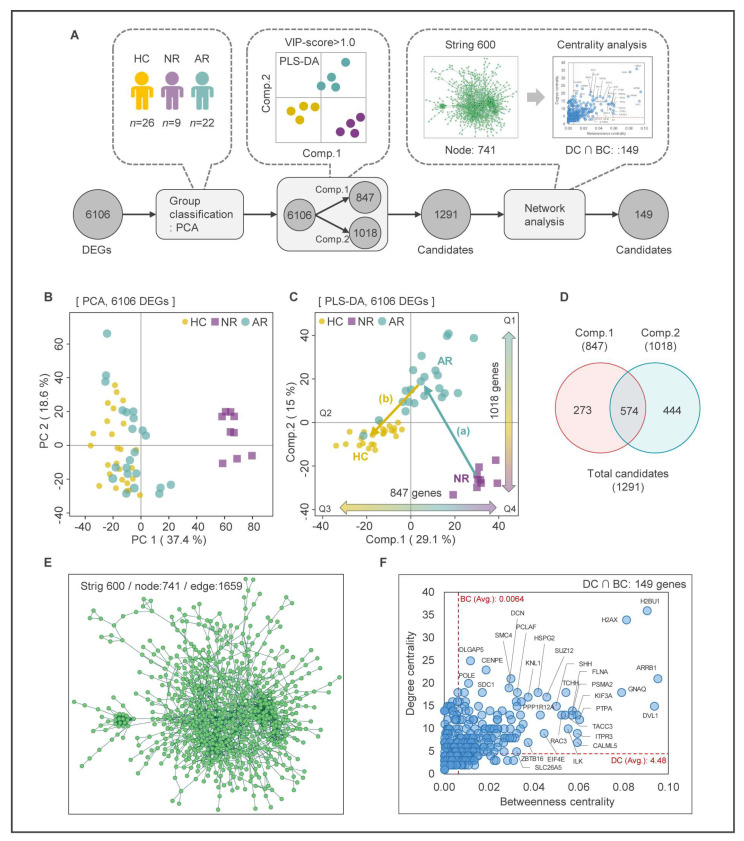
Integrative analysis of PLS-DA and PPI network analysis. (**A**) Flow chart of the integrative bioinformatics analysis. (**B**) PCA of 6106 DEGs from new classified groups (NR, AR, and HC). (**C**) PLS-DA score plot of the first two principal components for 6106 DEGs from samples of MUD patients (NR and AR) and HC. (**D**) Venn diagram of DEGs with a VIP score of 1.0 or higher for each component in the PLS-DA model. (**E**) PPI network of 1291 DEGs using STRING and Cytoscape. The network nodes represent proteins, and edges indicate the predicted functional associations between nodes. (**F**) Network centrality analysis for screening of hub genes (DC = 4.48 and BC = 0.00064). DC: degree centrality; BC: betweenness centrality.

**Figure 4 ijms-24-08672-f004:**
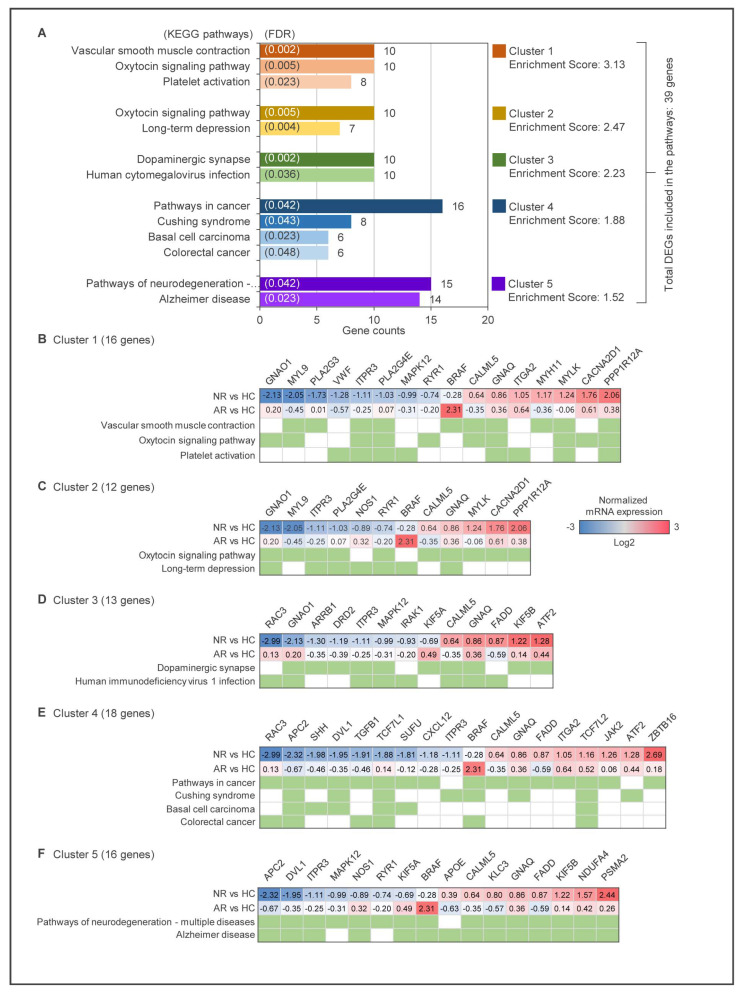
Functional annotation clustering analysis. (**A**) Functional annotation clustering analysis of 149 DEGs using KEGG pathways enrichment analysis in the DAVID. The representative groups with an enrichment score 1.5 or above are presented. The representative groups with a similarity term overlap of 5 or more are formed as clusters. In each cluster, KEGG pathway categories with an FDR of 0.05 or less are represented. The *x*-axis represents the number of genes, while the *y*-axis represents the KEGG pathway categories. (**B–F**) The DEGs included in the KEGG pathways of each cluster and their expression patterns. The green rectangles indicate the inclusion of DEGs in each KEGG pathway.

**Figure 5 ijms-24-08672-f005:**
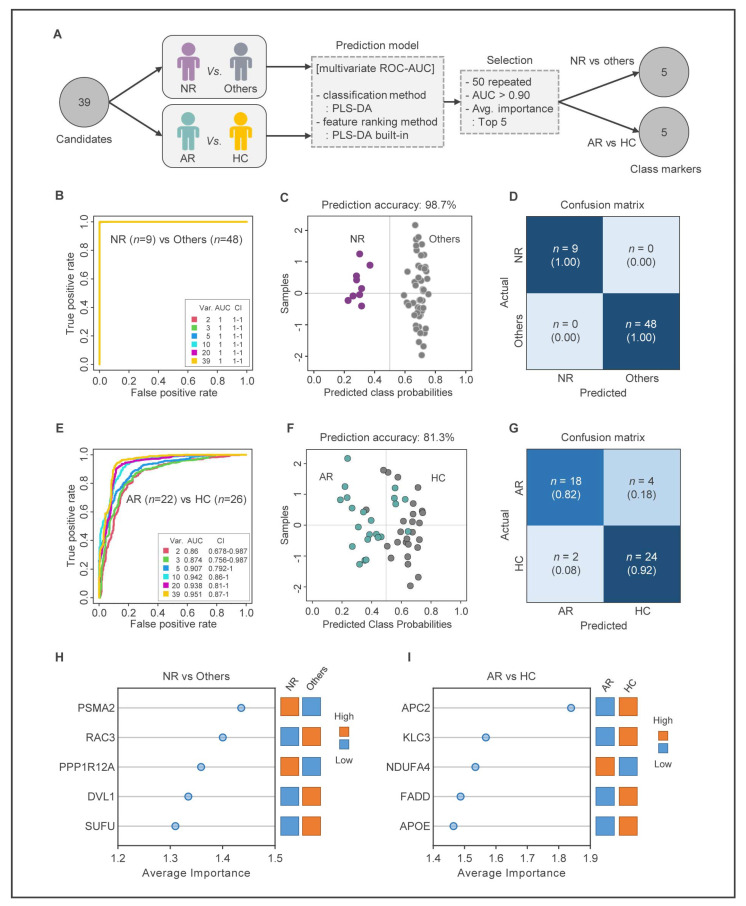
Construction of MUD prediction model using potential biomarkers. (**A**) Flow chart of multivariate ROC curve analysis and parameter optimization for screening of potential biomarkers. (**B**) The ROC curves and 5-fold cross-validation (CV) AUC values of 39 DEGs set for the classification of NR and Others (AR + HC). (**C**) The predicted class probabilities (average of the cross-validation) for each sample of NR (*n* = 9, purple dots) and others (*n* = 48, gray dots) groups were calculated using the best classifier (based on AUC). The classification boundary is always centered (x = 0.5) by using a balanced subsampling approach to model training. (**D**) The confusion matrix of the 39 DEGs set for the classification of NR and Others (AR + HC). (**E**) The ROC curves and 5-fold cross-validation (CV) AUC values of 39 DEGs set for the classification of AR and HC. (**F**) The predicted class probabilities for each sample of AR (*n* = 22, green dots) and HC (*n* = 26, gray dots) groups were calculated using the best classifier. (**G**) The confusion matrix of the 39 DEGs set for the classification of AR and HC. (**H**,**I**) Top 5 genes ranked by mean importance measure for the ROC curve analysis.

**Figure 6 ijms-24-08672-f006:**
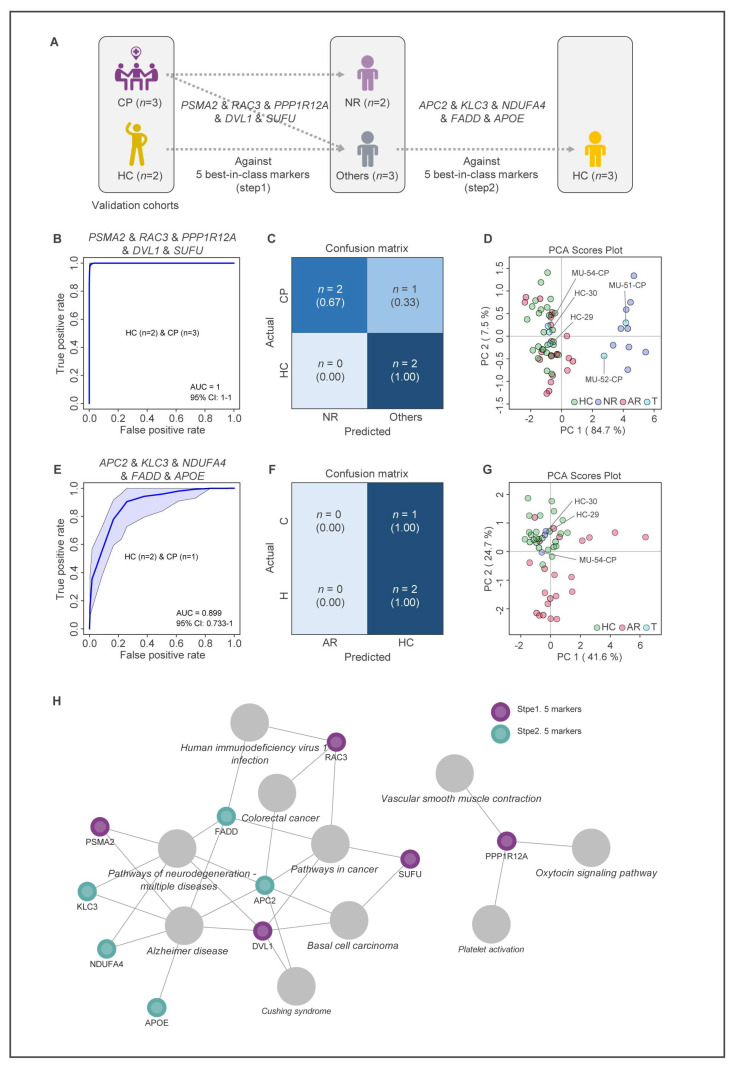
Validation of MUD prediction model. (**A**) Flow chart of validation strategy for two-step MUD prediction model. Combination of biomarker were applied in two steps of validation. (**B**) Validation of the first step of prediction model using 5 biomarkers (AUC = 1). (**C**) The confusion matrix of the first step validation. (**D**) PCA plot for prediction of validation sample (*n* = 5) from all samples (*n* = 62). (**E**) Validation of the second step of prediction model using 5 biomarkers (AUC = 0.891). (**F**) The confusion matrix of the second step validation. The shadow represents the 95% confidence interval. (**G**) PCA plot for prediction of validation sample (*n* = 3) from all samples (*n* = 51). (**H**) A hair follicle gene regulatory network associated with MUD.

## Data Availability

The RNA-seq data have been deposited to the Gene Expression Omnibus (GEO) database under the accession number GSE227204.

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
