# Peer review of "Identification of Potential Biomarkers for Diagnosis of Patients with Methamphetamine Use Disorder"

_ijms, 2023, doi:10.3390/ijms24108672_

Round 1
Reviewer 1 Report
This paper aimed to conduct a study of hair follicle cells from healthy controls, former MUD patients, and current MUD patients to identify potential non-invasive biomarkers. However, I would like to draw the attention of the authors to the important points that need to be corrected in the article. Before acceptance, the following points must be incorporated.1. In the abstract, it is required to focus on the contribution of the paper with respect to the other studies in the literature.
2. Give the key quantitative results at the end of the abstract.
3. The contribution is unclear. Refer to the following paper, and check how to specifically write the contribution at the end of the Introduction section. “A novel framework for prognostic factors identification of malignant mesothelioma through association rule mining”.
4. Clearly mention the research rap using the latest research.
5. Authors should add the literature review section and also add the latest literature; Investigation of miR-222 as a potential biomarker in diagnosis of patients with methamphetamine abuse disorder; Early Prediction of Malignant Mesothelioma: An Approach towards Non-invasive Method; A comprehensive study to delineate the role of an extracellular vesicle‐associated microRNA‐29a in chronic methamphetamine use disorder; A machine learning approach for identification of malignant mesothelioma etiological factors in an imbalanced dataset
6. It is also recommended to explain inclusion and exclusion criteria for the sampling of patients.
7. It is better to summarize the major findings of the paper within one-two sentences with experimental results.
8. Limitations of the proposed approach should be mentioned as a separate section along with the future directions.
9. There are some spelling and grammatical errors in the article. The entire article needs to be checked for spelling.
10. Authors should use some hypothesis testing to prove the results.
11. Overall speaking, the innovation points and main contributions of this paper need to be carefully reconsidered, and the innovation points should be presented more clearly and more prominent in terms of word expression and methodology & experiment design.
Reviewer 2 Report
-Please provide additional information about the biological measures that are proposed here and their relevance for predicting MUD.
-Please provide additional information about the multivariate measures that did not provide adequate fit to the data and require improvement. A better understanding of what these measures were would facilitate a much better understanding of the contribution made by the use of the new measures proposed here.
-Please provide sociodemographic information about the participants in all three groups. Did they differ based on characteristics like race, age, gender, and socioeconomic status, etc.? If so, that could contribute to the observed results.
-Do the genetic differences described in the results refer to epigenetic processes (DEGs and URGs)? If so, please just state this explicitly so it is clear that these are epigenetic differences that are being measured.
Round 2
Reviewer 1 Report
Accepted.